# NEXTBESTPATH: EFFICIENT 3D MAPPING OF UNSEEN ENVIRONMENTS

**Shiyao Li**[1]**, Antoine Guédon**[1]**, Clémentin Boittiaux**[1]**, Shizhe Chen**[2]**, Vincent Lepetit**[1]

[1]LIGM, École Nationale des Ponts et Chaussees, IP Paris, Univ Gustave Eiffel, CNRS, France
`{firstname.lastname}@enpc.fr`
[2]Inria, École normale supérieure, CNRS, PSL Research University, France
`{firstname.lastname}@inria.fr`

## ABSTRACT

This work addresses the problem of active 3D mapping, where an agent must find an efficient trajectory to exhaustively reconstruct a new scene. Previous approaches mainly predict the next best view near the agent's location, which is prone to getting stuck in local areas. Additionally, existing indoor datasets are insufficient due to limited geometric complexity and inaccurate ground truth meshes. To overcome these limitations, we introduce a novel dataset AiMDoom with a map generator for the Doom video game, enabling to better benchmark active 3D mapping in diverse indoor environments. Moreover, we propose a new method we call next-best-path (NBP), which predicts long-term goals rather than focusing solely on short-sighted views. The model jointly predicts accumulated surface coverage gains for long-term goals and obstacle maps, allowing it to efficiently plan optimal paths with a unified model. By leveraging online data collection, data augmentation and curriculum learning, NBP significantly outperforms state-of-the-art methods on both the existing MP3D dataset and our AiMDoom dataset, achieving more efficient mapping in indoor environments of varying complexity. Project page: `https://shiyao-li.github.io/nbp/`

## 1    INTRODUCTION

Autonomous 3D mapping of new scenes holds substantial importance for vision, robotics, and graphics communities, with applications including digital twins. In this paper, we focus on the problem of active 3D mapping, where the goal is for an agent to find the shortest possible trajectory to scan the entire surface of a new scene using a depth sensor.

This task is extremely challenging as the agent has to identify an efficient trajectory without knowing the scene in advance. Existing works can be broadly categorized into rule-based and learning-based approaches. Rule-based approaches, such as frontier-based exploration (FBE) (Yamauchi, 1997), utilize heuristic rules to select optimal frontiers at the boundaries of the already-known space for the next movement. Though being simple and generalizable, they fail to leverage data priors to develop more efficient planning strategies. To address this, learning-based methods, often referred to as next-best-view planning (NBV), train parametric policies for action prediction. Although NBV approaches have demonstrated promising results, most of them only are evaluated on single-object datasets or outdoor scenes (Guédon et al., 2022; Chang et al., 2015; Peralta et al., 2020), ignoring a critical but more difficult setting of indoor environments for active 3D mapping applications.

Existing indoor datasets (Xia et al., 2018; Chang et al., 2017), however, offer limited geometry complexity and often include imperfect ground truth meshes, making them inadequate to fully evaluate model performance in complex indoor environments. In this work, we automatically construct a new indoor dataset called AiMDoom for active 3D mapping. AiMDoom is built upon a map generator for the Doom video game, and features a wide range of indoor settings of four difficulty levels: Simple, Normal, Hard and Insane. As illustrated in Figure 1a, even in relatively simple indoor settings of our dataset, the state-of-the-art NBV approach MACARONS (Guédon et al., 2023) is frequently trapped in a limited area and misses substantial portions of the scene. This limitation arises because

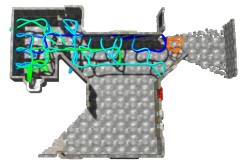 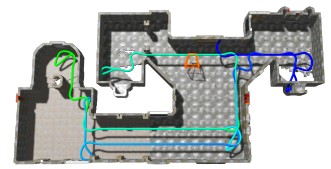 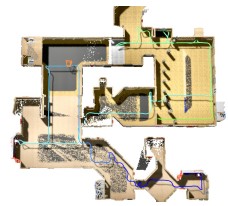

(a) MACARONS (simple scene).     (b) Our NBP (simple scene).     (c) Our NBP (hard scene).

Figure 1: Reconstruction results and trajectories of MACARONS (Guédon et al., 2023) and our NBP model. Guédon et al. (2023) fails to fully map the environment in simple scenes (a), while our NBP model manages to capture the full scene (b), even in much more complex geometry (c).

most NBV methods only look one step ahead to identify the next best view in neighbouring regions, making it difficult to explore under-reconstructed areas at far distances.

Some recent works (Chen et al., 2024; Feng et al., 2024; Zhan et al., 2022; Georgakis et al., 2022) attempt to overcome this limitation by searching for the next optimal view across a broader range. For example, Georgakis et al. (2022) utilizes a strategy that relies on averaging predicted uncertainties at each point along every sampled path, and uses a trained point-goal navigation model. However, training separate uncertainty map prediction and navigation models is less efficient, and the scene uncertainty does not directly align with the ultimate objective of 3D mapping.

Therefore, we further propose a novel approach called next-best-path (NBP) planning, which shifts from NBV approaches that predict a single nearby view, to predicting an optimal path in a unified model. Our model is composed of three key components: a mapping progress encoder, a coverage gain decoder and an obstacle map decoder. The mapping progress encoder efficiently encodes the currently reconstructed point cloud along with the agent's past trajectory. Based on the encoded representation, the coverage gain decoder predicts a value map over a large spatial range centred on the agent's current location. Each cell in the map represents the surface coverage gain accumulated along the optimal trajectory from the agent's location to the cell, which corresponds to the final metric for active mapping. The cell with the highest value score is viewed as a long-term goal. The obstacle map decoder predicts obstacles in both seen and unseen regions by leveraging the agent's current knowledge of the scene. This allows us to compute the shortest path to the long-term goal while avoiding obstacles. To train the model, we collect data online and iteratively improve the model. We also propose a data augmentation method that exploits a property of shortest paths and a combined curriculum and multitask learning strategy to enhance training efficiency.

We evaluate our methods on the existing indoor benchmark MP3D (Chang et al., 2017) and our dataset AiMDoom. The proposed NBP model significantly outperforms state-of-the-art methods on both datasets from simple (Figure 1b) to more complex indoor environments (Figure 1c).

Our key contributions can be summarized as follows:

• We introduce AiMDoom, the first benchmark to systematically evaluate active mapping in indoor scenes of different levels of difficulties.

• We propose a novel next-best-path approach that jointly predicts long-term goals with optimal reconstruction coverage gains, and obstacle maps for trajectory planning.

• Our approach achieved state-of-the-art results on both the AiMDoom and MP3D datasets.

## 2  RELATED WORK

**Active Mapping.** Active mapping aims to exhaustively reconstruct a 3D scene in the shortest possible time with a moving agent. Unlike SLAM (Chaplot et al., 2020; Placed et al., 2023; Matsuki et al., 2024), which addresses both localization and mapping, active mapping focuses on reconstruction, continuously selecting viewpoints to cover the entire scene, assuming the pose is known. Early methods often relied on frontier-based exploration (FBE) approaches (Yamauchi, 1997). The key idea is to move the agent toward a heuristically selected frontier along the boundary between reconstructed and unknown regions of the scene. Among different strategies (Bircher et al., 2016;

Table 1: **Comparison between AiMDoom and prior indoor 3D datasets.** Navigation complexity is the maximum ratio of geodesic to euclidean distances between any two navigable locations in the scene. Universal accessibility means whether windows and doors are accessible.

| Dataset | Replica | RoboTHOR | MP3D | Gibson (4+ only) | ScanNet | HM3D | AiMDoom (Ours) | | | |
|---|---|---|---|---|---|---|---|---|---|---|
| | | | | | | | Simple | Normal | Hard | Insane |
| Number of scenes | 18 | 75 | 90 | 571 (106) | 1613 | 1000 | 100 | 100 | 100 | 100 |
| Floor space (m$^2$) | 2.19k | 3.17k | 101.82k | 217.99k (17.74k) | 39.98k | 365.42k | 63.33k | 134.84k | 321.38k | 548.85k |
| Navigation complexity | 5.99 | 2.06 | 17.09 | 14.25 (11.90) | 3.78 | 13.31 | 11.31 | 18.38 | 36.05 | 45.25 |
| Universal accessibility | ✗ | ✗ | ✗ | ✗ | ✗ | ✗ | ✓ | ✓ | ✓ | ✓ |
| Easy expansion | ✗ | ✗ | ✗ | ✗ | ✗ | ✗ | ✓ | ✓ | ✓ | ✓ |

Cieslewski et al., 2017; Zhou et al., 2021; Tao et al., 2023) for frontier selection, moving to the nearest frontier serves as a strong baseline. Additionally, there are efforts (Cao et al., 2021; Xu et al., 2024) that combine global FBE and local planning strategies within a hierarchical optimization framework to enhance exploration. However, these FBE-based approaches are heuristic-based and cannot exploit prior learned from data to explore more efficiently, restricting their performance in complex environments.

To address this limitation, learning-based approaches have been explored to select the next-best views (NBV) for efficient 3D mapping. The NBV-based methods train models to select the optimal pose from nearby camera poses (Guédon et al., 2022; 2023; Lee et al., 2023) or from a limited predefined view space such as a hemisphere (Zhan et al., 2022; Lee et al., 2022; Peralta et al., 2020; Zeng et al., 2020; Mendoza et al., 2020). While these methods show promising results to reconstruct single objects, their performance remains limited in large environments. Due to the narrow search space for the next pose, NBV methods behave like a greedy policy and thus can easily get stuck in local regions. To mitigate this, some works Ramrakhya et al. (2022); Chen et al. (2023) use imitation learning to learn from human demonstrates which prioritize unseen exploration but with the cost of heavy labelling. More recently, efforts have been made to enlarge the search range for the next best view (Chen et al., 2024; Ran et al., 2023; Pan et al., 2022; Georgakis et al., 2022). However, these methods are still primarily evaluated on single-object datasets with small moving steps, and often rely on optimizing indirect metrics like reconstruction uncertainty (Georgakis et al., 2022), which are not directly aligned with the goal of exhaustive 3D reconstruction. In this work, we extend the evaluation to more complex indoor environments and also introduce a new surface coverage gain criterion that optimizes the coverage gain along the best trajectory towards a long-term goal.

**3D mapping datasets.** Existing datasets for 3D mapping mainly focus on single isolated objects such as those in ShapeNet (Chang et al., 2015) and OmniObject3D (Wu et al., 2023), or outdoor scenes (Lu et al., 2023; Hardouin et al., 2020), where the agent only needs to move around the scene to achieve full reconstruction. These datasets are comparatively less complex than indoor environments where the agent must enter into the scene. The indoor scenes contain unique challenges such as dead ends and tight corners, which often force the agent to backtrack without significantly improving its objective.

While some works (Yan et al., 2023; Georgakis et al., 2022; Ramakrishnan et al., 2020) incorporate indoor scene datasets such as Gibson (Xia et al., 2018) and MP3D (Chang et al., 2017), these often exhibit significant limitations. Existing synthetic datasets (Straub et al., 2019; Deitke et al., 2020) often lack scene complexity, whereas real-world scans (Dai et al., 2017; Ramakrishnan et al., 2021), despite offering greater representational fidelity, are constrained by limited structural and map diversity and often suffer from substantial noise artifacts. This lack of reliable datasets prevents comprehensive evaluation in active 3D mapping tasks. In this work, we propose a new dataset - AiMDoom, designed for benchmarking active mapping in indoor environments of different complexities.

## 3 THE AiMDoom Dataset

In this section, we introduce **AiMDoom**, a new dataset for **A**ctive 3D **M**apping in complex indoor environments based on the **Doom** video game [1]. As Doom features a wide variety of indoor settings, we use its map generator to create four sets of maps of increasing geometric complexity: Simple,

---

[1] https://en.wikipedia.org/wiki/Doom_(franchise)

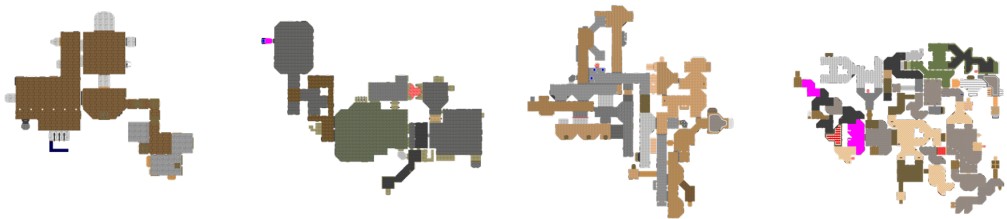

(a) Bird-eye views of samples from the ***Simple, Normal, Hard, and Insane*** levels (from left to right).

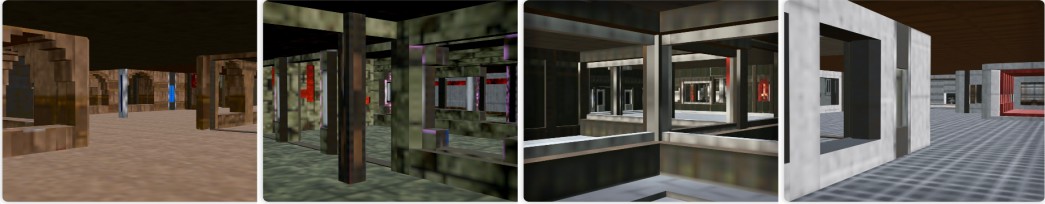

(b) Representative images showing the internal structural composition of the scene.

Figure 2: **Maps from our AiMDoom dataset.** The AiMDoom dataset includes four levels of geometric complexity with various textures.

Normal, Hard, and Insane. In the following, we first detail how we built these maps and then discuss the key challenges presented in our AiMDoom dataset.

**Dataset construction.** We used the open-source software Obsidian [2] to automatically generate Doom maps as our indoor environments. Four sets of hyperparameters are proposed to control architectural complexity and texture styles in Obsidian. By varying these hyperparameters, we produced maps categorized into Simple, Normal, Hard and Insane difficulty levels. Each difficulty level is made of 100 maps with 70 for training and 30 for evaluation.

The maps include doors and windows, all of which are configured to be open. This allows the agent to see and pass through the doors and windows. We converted the maps to the widely used OBJ format, and used Blender (Community, 2018) to consolidate the texture images of each map into a single texture image. This makes the maps compatible with Pytorch3D (Ravi et al., 2020) and Open3D (Zhou et al., 2018). Further details are presented in the supplementary material.

**Key challenges.** The AiMDoom dataset presents three key challenges for active 3D mapping. Firstly, the dataset features environments with intricate geometries and layouts as shown in Figure 2, making it challenging to determine the optimal exploration direction for effective mapping. Secondly, the maps have small doors and narrow corridors, requiring careful path planning to navigate. Finally, the map diversity requires the reconstruction system to generalize across different scenes. Table 1 compares AiMDoom with existing indoor 3D datasets (Straub et al., 2019; Deitke et al., 2020; Chang et al., 2017; Dai et al., 2017; Xia et al., 2018; Ramakrishnan et al., 2021), highlighting our dataset's strengths in scene area and navigation complexity.

We will release the dataset along with a comprehensive toolkit to generate the data, which enables easy expansion of the dataset for future research.

## 4 LEARNING ACTIVE 3D MAPPING

### 4.1 OVERVIEW

**Problem definition.** Active 3D mapping aims to control an agent, such as an unmanned aerial vehicle (UAV) or wheeled robot, to efficiently and exhaustively reconstruct a 3D scene. The agent starts at a random location within the scene, and at each time step $t$, it receives an RGB-D image $I_t$ and must predict the next one $c_t = (c_t^{\text{pos}}, c_t^{\text{rot}})$ in the immediate surrounding of the agent. Here, $c_t^{\text{pos}}$ denotes the position coordinates, and $c_t^{\text{rot}}$ represents the orientation angles. The agent continually

---

[2]https://obsidian-level-maker.github.io/

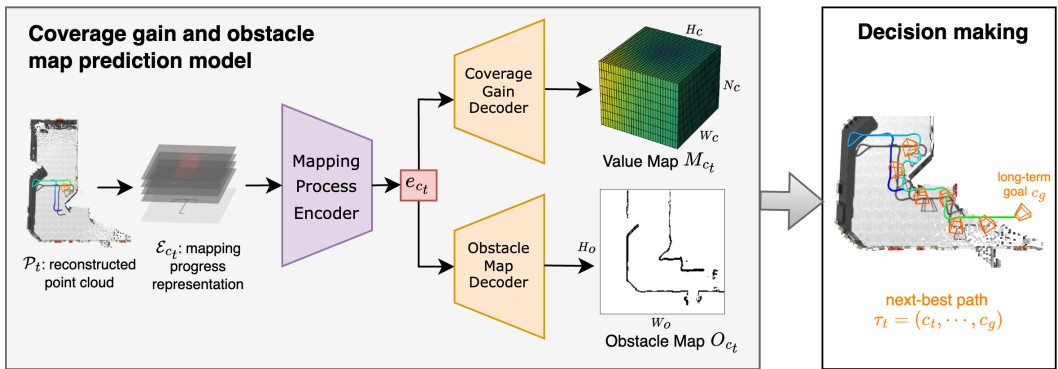

Figure 3: **Overview of the proposed next-best-path (NBP) framework.** The model (left, see Section 4.2) predicts a value map of coverage gain and an obstacle map, which are used for decision making (right, see Section 4.3) to obtain a next-best path.

predicts successive $c_t$ until a predefined time limit $T$ is reached. The final output is the reconstructed 3D point cloud of the explored environment.

**Overview of our approach.** Existing approaches for active mapping (Guédon et al., 2022; 2023) typically predict the next camera pose $c_t$ in a greedy manner, which often suffers from getting stuck in limited areas. To address this limitation, we propose a novel approach that predicts a long-term goal camera pose and uses it to guide the next camera pose selection. Given all past observations and camera poses, our model predicts two key components centred on the agent's current pose $c_t$: (1) a value map $M_{c_t}$, which estimates the surface coverage gain of candidate poses $c$ in the surrounding of $c_t$, and (2) an obstacle map $O_{c_t}$, which accounts for both visible and predicted unseen obstacles in the environment. From the value map $M_{c_t}$, we derive the long-term goal pose $c_g$ and combine it with the obstacle map $O_{c_t}$ to compute an optimal path $\tau_t = (c_t, c_{t+1}, \cdots, c_g)$ that navigates the agent from its current pose $c_t$ to the goal pose $c_g$. This long-term goal-driven strategy helps the model avoid the pitfalls of short-sighted decisions and enhances coverage efficiency.

In the following, we first describe the model for $M_{c_t}$ and $O_{c_t}$ prediction in Section 4.2, then followed by the decision-making process to determine the next best path $\tau_t$ in Section 4.3. Finally, in Section 4.4, we introduce the training algorithm for our model.

## 4.2 COVERAGE GAIN AND OBSTACLE PREDICTION MODEL

Figure 3 depicts the deep model we use to predict the coverage gains and the obstacle map. We detail this model below.

**Mapping Progress Encoder.** Let's denote $\mathcal{P}_t$ the reconstructed point cloud at each time step $t$, obtained by adding the back-projected depth image $I_t$ to the previously accumulated point cloud $\mathcal{P}_{t-1}$. Directly encoding the point cloud via 3D neural networks can be complex and inefficient. Therefore, we convert the 3D point cloud into multiple 2D images as inputs to a 2D-based encoder.

To be specific, we first centre and crop the point cloud based on the agent's current position $c_t$. Centering the input on the agent makes the model invariant to the agent's position and thus improves generalization. Then, the point cloud is divided into $K$ horizontal layers along the gravity axis. For each layer, we average the occupancy value along the gravity axis to transform each 3D data into a 2D image. In this image, each pixel encodes the density of 3D points within a specific height range. The stack of $K$ point cloud projected images provides a simplified yet informative representation of the 3D structure.

Similarly, we project the 3D trajectory of the agent's past camera poses onto a 2D plane where each pixel denotes the frequency of visits to that location. This plane serves to mitigate the exploratory value of previously traversed regions. We define $\mathcal{E}_{c_t}$ to include the $K$ point cloud projected images and a single historical trajectory image.

Given the stacked 2D images of $\mathcal{E}_{c_t}$, we employ an Attention UNet (Oktay et al., 2018) encoder with 4 downsampling convolutional blocks to extract mapping progress features $e_{c_t}$.

**Coverage Gain Decoder.** This decoder predicts from $e_{c_t}$ a 3D value map $M_{c_t} \in \mathbb{R}^{H_c \times W_c \times N_c}$ centered on the agent. It is composed of two upsampling convolutional blocks with an attention mechanism. The first two dimensions of the predicted value map, $H_c$ and $W_c$, correspond to the camera's 2D position in the environment, while the third dimension $N_c$ represents different camera orientations. Each value in $M_{c_t}$ quantifies the estimated coverage gain achievable by moving the camera along the shortest trajectory from its current pose to the specific camera pose. The value map $M_{c_t}$ guides the selection of both long-term goal poses $c_g$ and intermediate poses along the trajectory, enabling a two-stage optimization for efficient exploration, which will be discussed in Section 4.3.

**Obstacle Map Decoder.** This decoder predicts the geometric layout $O_{c_t} \in \mathbb{R}^{H_o \times W_o}$ of the current moving plane, also from the encoder output $e_{c_t}$. $O_{c_t}$ is a binary map representing potential obstacles around the current agent location, which is used for path planning. To be noted, $O_{c_t}$ includes not only visible obstacles but also anticipated unseen obstacles based on the structure of the partially reconstructed point cloud, providing useful priors for navigation. This decoder is implemented using Attention U-Net with 4 upsampling convolutional blocks, and the output is passed through a sigmoid activation function to generate the binary obstacle map.

## 4.3 Decision Making for Next-best-path Prediction

We derive both a long-term goal $c_g$ and next-best-path $\tau_t = (c_t, c_{t+1}, \dots, c_g)$ from the predicted $M_{c_t}$ and $O_{c_t}$, employing different decision making strategies for training and inference. During training, we balance exploitation and exploration, while we prioritize exploitation during inference.

**Training phase.** We rely on the Boltzmann exploration strategy (Cesa-Bianchi et al., 2017) to sample a camera pose as the goal $c_g$ based on the value map $M_{c_t}$. The probability of selecting a camera pose $c$ as the goal is given by:

$$P(c_g = c) = \frac{\exp(M_{c_t}[c]/\beta)}{\sum_{c' \in C} \exp(M_{c_t}[c']/\beta)} \,, \tag{1}$$

where $C$ represents all possible camera poses within $M_{c_t}$, $\beta$ is the temperature parameter that balances exploration and exploitation, and $M_{c_t}[c]$ denotes the value of the cell for candidate $c$.

Once the long-term goal $c_g$ is sampled, we use the Dijkstra algorithm to find the shortest obstacle-free path from the current position $c_t^{\text{pos}}$ to goal position $c_g^{\text{pos}}$ with a ground truth obstacle map. To select camera orientation along the path, we also leverage $M_{c_t}$ to sample one orientation from $N_c$ potential orientations at each position. This strategy enhances data diversity and alleviates the risk of converging to local optima.

**Inference phase.** At inference, we take $c_g$ as the pose with the maximum value in $M_{c_t}$, and the path planning is based on the predicted obstacle map $O_{c_t}$ instead of ground truth. Each position in the trajectory is assigned the optimal orientation from the heatmap $M_{c_t}$ for its location. In practice, the predicted obstacle map may not be entirely accurate. Encountering an unexpected obstacle requires halting the trajectory and initiating a new decision-making phase.

## 4.4 Model Training

Algorithm 1 outlines the training procedure for our model. We first gather training data from all training scenes using the current model, and then update the model with the new data. This process is repeated iteratively until the model achieves convergence. We detail below the data collection, training objectives to update the model, and the training strategy.

---

**Algorithm 1** Training procedure.

$N$: number of training iterations
$N_e$: number of iterations using easy data
$S_n$: the number of trajectories per scene
Initialize memory $\mathcal{M} \leftarrow \emptyset$ and model parameters $\theta$
**for** $n \leftarrow 1$ **to** $N$ **do**
    Initialize training set $\mathcal{T} \leftarrow \emptyset$
    **for** each scene in training set **do**
        **for** $s \leftarrow 1$ **to** $S_n$ **do**
            Collect training data $\{d_l\}_{l=1}^L$
            **if** $n \leq N_e$ **then** $\mathcal{T} \leftarrow \mathcal{T} \cup \{d_l : t \geq 10\}_{l=1}^L$
            **else** $\mathcal{T} \leftarrow \mathcal{T} \cup \{d_l\}_{l=1}^L$ **endif**
        **end for**
    **end for**
    $\mathcal{M} \leftarrow \mathcal{M} \cup \mathcal{T}$
    $\mathcal{T} \leftarrow \mathcal{T} \cup \text{RandomSample}(\mathcal{M} \setminus \mathcal{T}, |\mathcal{T}|)$
    **for** $e \leftarrow 1$ **to** $E$ **do**
        Update $\theta$ with loss in Eq. (3) over $\mathcal{T}$
    **end for**
**end for**
**return** $\theta$

---

**Training data collection.** After sampling the goal pose $c_g$ and the trajectory $\tau_t$, we generate ground truth labels to train the value map $M_{c_t}$ and obstacle map $O_{c_t}$.

For $M_{c_t}$, we compute the coverage gain for the cell that corresponds to $c_g$ as the ground truth label. Let $\mathcal{P}_t$ and $\mathcal{P}_g$ denote the reconstructed point clouds at pose $c_t$ and $c_g$ respectively, where $\mathcal{P}_g$ is the result of accumulating depth information into $\mathcal{P}_t$ as the agent moves along the trajectory $\tau_t$. By comparing the reconstructed point clouds with the ground truth point cloud $\mathcal{P}^{\text{GT}}$, we can obtain the coverage gain $\Delta\text{Cov}_{c_t \to c_g}$:

$$\Delta\text{Cov}_{c_t \to c_g} = \frac{1}{N_{\text{GT}}} \sum_{i=1}^{N_{\text{GT}}} \left[ \mathbf{1}\left( \min_{y \in \mathcal{P}_g} \|x_i^{\text{GT}} - y\| < \epsilon \right) - \mathbf{1}\left( \min_{y \in \mathcal{P}_t} \|x_i^{\text{GT}} - y\| < \epsilon \right) \right] , \qquad (2)$$

where $N_{\text{GT}}$ is the number of points in $\mathcal{P}^{\text{GT}}$, $\|\cdot\|$ denotes the Euclidean distance, and $\varepsilon$ is a predefined distance threshold. Consequently, we set $\Delta\text{Cov}_{c_t \to c_g}$ as the ground truth value for $M_{c_t}[c_g]$.

For $O_{c_t}$, we use the 3D mesh of the scene to derive the ground truth obstacle map $O_{c_t}^{\text{GT}}$. Specifically, we intersect the 3D mesh with a plane at the agent's height, and project this intersection onto a 2D grid. This 2D grid is binarized to distinguish between obstacles and free space. Finally, we centre the 2D grid around the agent's current position as $O_{c_t}^{\text{GT}}$.

To enhance the efficiency of data generation, we further perform a data augmentation by leveraging the property of Dijkstra's algorithm, where every sub-path of a shortest path is also a shortest path. From a given path $\tau_t = (c_0 = c_t, \ldots, c_m = c_g)$, we compute the coverage gain $\Delta\text{Cov}_{c_i \to c_j}$ for each segment of the path $(c_i, c_j)$ where $0 \leq i < j \leq m$. More specifically, we update the ground truth values along the Dijkstra path $M_{c_i}^{\text{GT}}[c_j] = \Delta\text{Cov}_{c_i \to c_j}$. We also collect the input $\mathcal{E}_{c_i}$ and the ground truth of surrounding obstacles $O_{c_i}^{\text{GT}}$ for each $c_i \in \tau_t$. This significantly increases the number of training samples derived from a single trajectory.

We store all augmented pairs $\{d_l\}_{l=1}^L$, $d_l = (\mathcal{E}_{c_i}, M_{c_i}^{\text{GT}}, O_{c_i}^{\text{GT}})$ in memory for training, where $L$ is the length of the trajectory.

**Multi-task training.** We jointly train the coverage gain and obstacle map prediction using data stored in memory. We use the mean squared error (MSE) loss for training the coverage gain prediction, and the binary cross-entropy (BCE) loss for training the obstacle map prediction. To balance these two tasks effectively, we apply learnable uncertainty weights for each task, following Kendall et al. (2018). Our multi-task loss function for sample $d_l$ is formulated as follows:

$$\mathcal{L}(\theta; d_l) = \frac{1}{2\sigma_1^2} \mathcal{L}_{\text{MSE}}(M_{c_i}^{\text{GT}}, \hat{M}_{c_i}) + \frac{1}{\sigma_2^2} \mathcal{L}_{\text{BCE}}(O_{c_i}^{\text{GT}}, \hat{O}_{c_i}) + \log \sigma_1 + \log \sigma_2 , \qquad (3)$$

where $\theta$ represents the model parameters, $\sigma_1$ and $\sigma_2$ are learnable uncertainty weights, $\hat{M}_{c_i}$ and $\hat{O}_{c_i}$ are the model's predictions for the coverage gain and obstacle maps respectively.

**Training strategy.** We adopt a curriculum training strategy (Liu et al., 2017; Yuan et al., 2022; Yan et al., 2021; De Lange et al., 2021) to train our model, starting with easier-to-predict samples and gradually incorporating the entire dataset. In particular, we consider that the initial steps of a trajectory are more challenging since the agent has limited observations. Therefore, during the first $N_e$ training iterations, we exclude samples from the first 10 steps in a trajectory. After $N_e$ iterations, all samples in a trajectory are used in training.

During each training iteration, we use a balanced combination of previously stored data from the memory and newly collected data generated by the current model (Wulfmeier et al., 2018; Mnih, 2013; Rolnick et al., 2019; Aljundi et al., 2019), which helps prevent catastrophic forgetting. Each training phase is limited to $E$ epochs to balance between enhancing performance and preventing overfitting on sub-optimal data.

## 5 EXPERIMENTS

### 5.1 EXPERIMENTAL SETUP

**Dataset and simulation setup.** We evaluate our model on the Matterport3D (MP3D) dataset (Chang et al., 2017) and our own AiMDoom dataset.

Table 2: **Evaluation results on AiMDoom Dataset.** For each difficulty level, all baseline models, including ours, are trained from scratch on the corresponding training set to ensure a fair comparison.

| | Simple | | Normal | | Hard | | Insane | |
|---|---|---|---|---|---|---|---|---|
| | Final Cov. | AUCs | Final Cov. | AUCs | Final Cov. | AUCs | Final Cov. | AUCs |
| Random | $0.323_{\pm0.156}$ | $0.270_{\pm0.135}$ | $0.190_{\pm0.124}$ | $0.152_{\pm0.103}$ | $0.124_{\pm0.082}$ | $0.088_{\pm0.060}$ | $0.074_{\pm0.048}$ | $0.050_{\pm0.035}$ |
| FBE | $0.760_{\pm0.174}$ | $0.605_{\pm0.171}$ | $0.565_{\pm0.139}$ | $0.415_{\pm0.109}$ | $0.425_{\pm0.114}$ | $0.311_{\pm0.080}$ | $0.330_{\pm0.097}$ | $0.239_{\pm0.079}$ |
| SCONE | $0.577_{\pm0.173}$ | $0.483_{\pm0.138}$ | $0.412_{\pm0.114}$ | $0.313_{\pm0.087}$ | $0.290_{\pm0.093}$ | $0.210_{\pm0.072}$ | $0.196_{\pm0.079}$ | $0.140_{\pm0.060}$ |
| MACARONS | $0.599_{\pm0.200}$ | $0.479_{\pm0.172}$ | $0.418_{\pm0.120}$ | $0.314_{\pm0.088}$ | $0.302_{\pm0.097}$ | $0.218_{\pm0.070}$ | $0.192_{\pm0.078}$ | $0.139_{\pm0.058}$ |
| **NBP (Ours)** | $\mathbf{0.879}_{\pm0.142}$ | $\mathbf{0.692}_{\pm0.156}$ | $\mathbf{0.734}_{\pm0.142}$ | $\mathbf{0.526}_{\pm0.112}$ | $\mathbf{0.618}_{\pm0.153}$ | $\mathbf{0.432}_{\pm0.115}$ | $\mathbf{0.472}_{\pm0.095}$ | $\mathbf{0.312}_{\pm0.073}$ |

For MP3D, we use the same setting as prior work (Yan et al., 2023) for fair comparison. The input posed depth images have a resolution of $256 \times 256$ with a horizontal field of view (hFOV) of $90°$. The mobile agent starts in the traversable space at a height of $1.25m$ and chooses its next camera pose by moving forward by $6.5cm$ or turning left/right by $10°$. Depending on the size of each scene, the agent can take a maximum of 1000 or 2000 steps. We focus only on single-floor scenes following Yan et al. (2023) with 10 and 5 scenes in training and evaluation respectively.

For AiMDoom, we utilize a 70/30 train/test split for scenes in each difficulty level. The input RGB-D images are rendered at the resolution of $456 \times 256$ with a vFOV of $60°$. The agent navigates in a traversable space of height $1.65m$. The moving step includes 4 position movements (move forward, backward, left, or right by $1.5m$) and 8 rotation movements (turn left or right by increments of $45°$, covering the full $360°$). For dense reconstruction, all methods capture three additional images between adjacent poses using linear interpolation. The maximum steps for Simple, Normal, Hard, and Insane levels are set to 100, 200, 400, and 500 respectively, to adapt to their different complexity.

**Evaluation metrics.** We follow prior works (Chen et al., 2024; Guédon et al., 2023) and adopt two key metrics to evaluate the performance of active 3D reconstruction: (1) ***Final Coverage*** measures the scene coverage at the end of the trajectory, and (2) ***AUCs*** evaluates the efficiency of the reconstruction process by calculating the area under the curve of coverage over time. The surface coverage is computed using ground truth meshes, consistent with prior work (Guédon et al., 2023). We evaluate five trajectories per scene using identical random initial camera poses for different methods. We report the mean and standard deviation for each metric across all testing trajectories.

For a fair comparison with prior work in MP3D, we employ another set of metrics to evaluate coverage: (1) ***Comp. (%)***, the proportion of ground truth vertices within $5cm$ of any observation, and (2) ***Comp. (cm)***, the average minimum distance between ground truth vertices and observations.

**Implementation details.** Our model takes a stack of $K = 4$ projected 2D images and one previous trajectory projected image as inputs, each with a resolution of $256 \times 256$ covering a $40m \times 40m$ exploration area centred on the camera's current position. The extracted feature $e_{c_t}$ from the encoder is of size $16 \times 16 \times 1024$. The output value map $M_{c_t}$ is of size $64 \times 64 \times 8$ and an obstacle map of $256 \times 256 \times 1$, both representing the same $40m \times 40m$ area. The model is trained for at most $N = 15$ iterations, with the first $N_e = 1$ iterations using easier samples and $S_n = 2$ trajectories per scene. For subsequent iterations, we use all samples and reduce the trajectory count to $S_n = 1$ per scene. Each trajectory has a length of 100 steps and starts at a random location. During the first data collection iteration, we randomly sample 1,000 validation examples from memory and exclude them from training. Gradient accumulation is used in training which results in an effective batch size of 448. The learning rate is set to 0.001 and is decayed by a factor of 0.1 if the validation loss plateaus. We apply early stopping to terminate training when validation loss no longer decreases. The training is performed on a single NVIDIA RTX A6000 GPU, with an average completion time of 25 hours.

## 5.2 COMPARISON WITH STATE OF THE ART METHODS

**MP3D.** We compare our method with five baselines on the MP3D dataset, including: 1) *Random*, which randomly selects a camera pose among all candidates for the next step; 2) *Frontier-based Exploration (FBE) (Yamauchi, 1997)*, which heuristically moves the agent to the nearest frontier; 3) *OccAnt (Ramakrishnan et al., 2020)*, which predicts the occupancy status of unexplored areas and rewards the agent for accurate predictions; 4) *UPEN (Georgakis et al., 2022)*, which utilizes

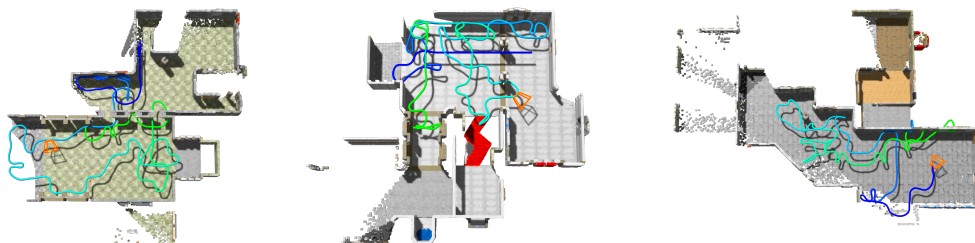

(a) Results of MACARONS. It generates complicated trajectories and often gets trapped in local areas.

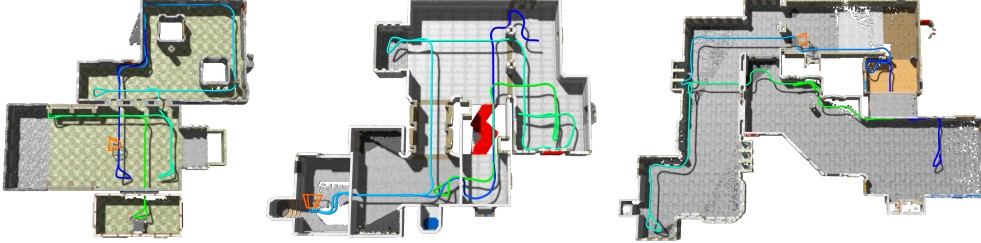

(b) Results of our NBP method. It efficiently travels in the scene and reconstructs the scene well.

Figure 4: **Comparison of our NBP method with the state-of-the-art MACARONS method.** Both methods start from the same initial pose, marked in deep blue. We also include a demonstration video of active mapping using our method in the supplementary materials.

an ensemble of occupancy prediction models to guide the agent towards paths with the highest uncertainty; 5) *ANM (Yan et al., 2023)*, which guides exploration through a continually-learned neural scene representation. Results in Table 3 show our NBP performs best, with a 6.23 absolute gain for the completion ratio compared to the state-of-the-art ANM (Yan et al., 2023) model.

Table 3: Comparison on the MP3D dataset.

| Method | Comp. (%) ↑ | Comp. (cm) ↓ |
|---|---|---|
| Random | 45.67 | 26.53 |
| FBE | 71.18 | 9.78 |
| UPEN | 69.06 | 10.60 |
| OccAnt | 71.72 | 9.40 |
| ANM | 73.15 | 9.11 |
| **NBP (ours)** | **79.38** | **6.78** |

**AiMDoom.** The proposed AiMDoom dataset is more challenging than MP3D dataset for active 3D mapping. We benchmark our approach against state-of-the-art Next-Best-View (NBV) approaches, including: *SCONE (Guédon et al., 2022)* which employs volumetric integration to sum the potential visibility points for each candidate camera pose in the subsequent step and is trained using supervised learning; and *MAC-ARONS (Guédon et al., 2023)* which quantifies the coverage gains of potential next camera poses to select the best one and utilizes a self-supervised online learning paradigm. Both approaches select the next camera pose in a greedy manner. Unfortunately, we were unable to include UPEN (Georgakis et al., 2022) and ANM (Yan et al., 2023) in our comparison. These methods rely on the navigation policy DD-PPO (Wijmans et al., 2019) trained on their environments (Savva et al., 2019), which requires extensive GPU hours and thus is infeasible to retrain it on our dataset. However, we implemented FBE (Yamauchi, 1997) on our dataset, a recognized strong baseline in reconstruction and exploration tasks.

As shown in Table 2, our method significantly outperforms the baselines across all metrics on four levels of AiMDoom. While NBV approaches such as SCONE and MACARONS excel in outdoor or single-object scenarios, their performance deteriorates in complex indoor environments. As illustrated in Figure 4, MACARONS struggles to escape local areas due to its short-term focus. It only selects the next best pose in nearby regions, and once these areas - such as the interior of a single room - are fully reconstructed, it has difficulty moving out of the room to explore under-explored, distant regions. In contrast, our approach overcomes this limitation by incorporating long-term goal guidance to determine the next-best path. In addition, our method surpasses the strong baseline FBE. Although FBE enables better exploration compared to state-of-the-art NBV methods on our dataset, its simple heuristic of moving to the nearest frontier leads to sub-optimal scene reconstruction as it lacks strategic planning for efficient coverage.

Despite the superior performance of our model, the results in hard and insane environments are still unsatisfactory, highlighting the significant challenges posed in our dataset.

## 5.3 ABLATION STUDY

In this section, we perform ablation experiments to demonstrate the effectiveness of different components in our model. All the experiments below are conducted on the Normal level of AiMDoom.

**Spatial range of long-term goal.** We compare the impact of different spatial ranges for the prediction of the value map $M_{c_t}$ and obstacle map $O_{c_t}$, which in turn determines the maximum distance of the long-term goals $c_g$. Specifically, we experiment with map sizes of $20m \times 20m$ to $50m \times 50m$. The results are presented in Figure 5. When the value map covers a smaller area, the goal $c_g$ is close to the agent's current position, leading to behaviour similar to existing NBV methods that struggle with explo-

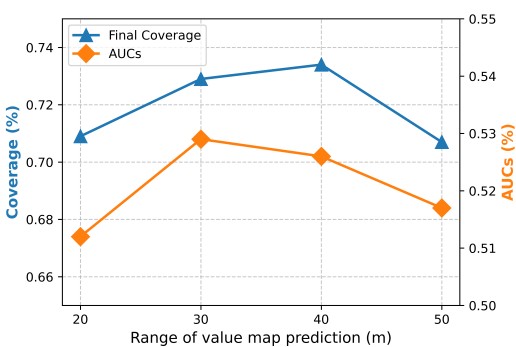

Figure 5: Comparisons of different spatial ranges for value map prediction.

ration. On the other hand, if the map size is too large, predicting $M_{c_t}$ and $O_{c_t}$ becomes much more challenging. Our findings demonstrate that selecting an appropriate spatial range for the value map is crucial for balancing exploration efficiency and prediction accuracy.

**Oracle obstacle map.** In Table 4, we replace the predicted obstacle map with the ground truth obstacle map for path planning during inference, while maintaining to use the predicted value map for long-term goals. Using the oracle obstacle map improves the performance by 0.074 on final coverage and 0.054 on AUCs, but is far from perfect. This suggests that the major bottleneck is the value map prediction.

Table 4: Ablation study on using the oracle map for obstacle avoidance at inference.

| Obstacle Map | Final Cov. | AUCs |
|---|---|---|
| Predicted | 0.734 ±0.142 | 0.526 ±0.112 |
| Oracle | **0.808** ±0.115 | **0.580** ±0.105 |

Table 5: Comparison of single-task and multi-task learning for the value map and obstacle map prediction.

| Strategy | Final Cov. | AUCs |
|---|---|---|
| Single-task | 0.712 ±0.136 | 0.501 ±0.101 |
| Multi-task | **0.734** ±0.142 | **0.526** ±0.112 |

**Multi-task training.** We also explore the influence of multi-task learning in predicting the value map $M_{c_t}$ and the obstacle map $O_{c_t}$. For comparison, we train two separate models that use the same input to predict $M_{c_t}$ and $O_{c_t}$ respectively. The results show that multi-task learning improved the precision of obstacle prediction to 0.805, exceeding the 0.754 achieved by single-task learning. Table 5 further demonstrates that multi-task learning achieves better performance, indicating that the two tasks complement each other to enhance learning.

## 6 CONCLUSION

In this paper, we tackle the challenging problem of active 3D mapping of unknown environments. We introduce a new dataset, AiMDoom, designed to benchmark active mapping in indoor scenes with four difficulty levels. Our evaluations of existing methods on the AiMDoom dataset reveal shortcomings of short-sighted next-best-view prediction in complex large indoor environments. Hence, we propose the next-best-path (NBP) method, which integrates a mapping progress encoder, a coverage gain decoder and an obstacle map decoder. The NBP model can efficiently reconstruct unseen environments guided by predicted long-term goals, achieving state-of-the-art performance on both the MP3D and AiMDoom datasets. However, we observe considerable room for improvement in more difficult levels of our dataset, and the major limitation lies in long-term goal prediction rather than obstacle map prediction.

## 7 ACKNOWLEDGEMENTS

This project was funded in part by the European Union (ERC Advanced Grant explorer Funding ID #101097259). This work was granted access to the HPC resources of IDRIS under the allocation 2024-AD011014703R1 made by GENCI. We thank Ruijie Wu for his valuable contribution to mesh editing throughout the AiMDoom dataset generation.

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

# APPENDIX

## A  DATASET

**Dataset construction.** To ensure that each map offers full accessibility for various robotic platforms such as unmanned aerial vehicles (UAVs) and wheeled robots, we configure all doors and windows to remain open during map generation. However, we observe that Obsidian does not consistently guarantee accessibility to all areas. To resolve this, we manually edited each scene to ensure the traversability of windows, doors, and hidden passages.

**Dataset statistic.** To calculate navigation complexity, we sampled location pairs from four environments ranging in difficulty from simple to insane, at rates of 1%, 0.1%, 0.05%, and 0.02%, constrained by computational resource limitations. Consequently, our data effectively represents a lower bound of navigation complexity. Despite this conservative sampling approach, our dataset remains the most complex one, offering a significant challenge for future research.

Figure 6 presents more examples of maps from our dataset, showcasing various levels and diverse scenarios.

## B  DETAILS OF MAPPING PROCESS ENCODER

We provide more details of the Mapping Process Encoder of our proposed approach in this section.

The mapping encoding is predicted from both the current reconstruction progress and historical trajectory data. At each time step $t \geq 0$, we construct and refine a surface point cloud $\mathcal{P}_t$ by integrating information from newly captured depth map $D_t : \Omega \to \mathbb{R}^+$ and merging it with our existing reconstructed point cloud. For each camera pose $c_t = (c_t^{\text{pos}}, c_t^{\text{rot}})$, we transform the corresponding depth map $D_t$ into a set of 3D points. This transformation makes use of the camera's intrinsic matrix $K \in \mathbb{R}^{3 \times 3}$ and the pose matrix $T_t \in SE(3)$, derived from the 6D pose $c_t$:

$$\mathbf{p}_{surface}(u,v) = T_t \cdot \left( D_t(u,v) \cdot K^{-1} \cdot [u \ v \ 1]^\top \right), \quad (u,v) \in \Omega, \tag{4}$$

where $\Omega \subset \mathbb{R}^2$ represents the domain of the depth map. We accumulate points over time:

$$\mathcal{P}_t = \mathcal{P}_{t-1} \cup \{ \mathbf{p}_{surface}(u,v) \mid (u,v) \in \Omega, D_t(u,v) > 0 \}. \tag{5}$$

To enhance scalability and generalization, we introduce a slice mapping approach that transforms the point cloud into a set of $K$ images. We begin by filtering the point cloud based on the camera's position:

$$\mathcal{P}_{c_t}^f = \{ \mathbf{p} = (p_x, p_y, p_z) \in \mathcal{P}_t \mid |p_x - x_{c_t}| \leq r \text{ and } |p_z - z_{c_t}| \leq r \}, \tag{6}$$

where $r$ is the radius of our observation window and $(x_{c_t}, y_{c_t}, z_{c_t})$ is the current camera position. We then divide $\mathcal{P}_{c_t}^f$ into $n$ equal vertical slices along the Y-axis, $y_{min}$ and $y_{mmax}$ come from a defined exploration bounding box, as Guédon et al. (2023) did:

$$\mathcal{S}_{c_t,j} = \{ \mathbf{p} = (p_x, p_y, p_z) \in \mathcal{P}_{c_t}^f \mid y_{min} + (j-1)h_{\text{slice}} \leq p_y < y_{\min} + jh_{\text{slice}} \}, \tag{7}$$

where $h_{\text{slice}} = (y_{\max} - y_{\min})/n$ and $j \in 1, \ldots, n$. Each slice $\mathcal{S}_{c_t,j}$ is mapped to an image $I_{c_t,j}$ of size $H \times W$ using a projection function $\phi : \mathbb{R}^3 \to \mathbb{R}^2$:

$$\phi(\mathbf{p}) = \left( \left\lfloor \frac{(p_x - x_{c_t} + r) \cdot W}{2r} \right\rfloor, \left\lfloor \frac{(p_z - z_{c_t} + r) \cdot H}{2r} \right\rfloor \right). \tag{8}$$

This projection centres the camera in the middle of the image. Finally, we calculate the point density for each pixel $(u,v)$ in the image $I_{c_t,j}$:

$$I_{c_t,j}(u,v) = \int_{\mathcal{S}_{c_t,j}} \delta(\phi(\mathbf{p}) - (u,v))d\mathbf{p}. \tag{9}$$

Here, $\delta(\cdot)$ is the Dirac delta function and $\mathbf{p}$ represents points from the slice $\mathcal{S}c_t, j$. This process yields $n$ density images $I_{c_t,1}, \ldots, I_{c_t,n}$ for each time step $t$, effectively transforming 3D point cloud data into 2D representations.

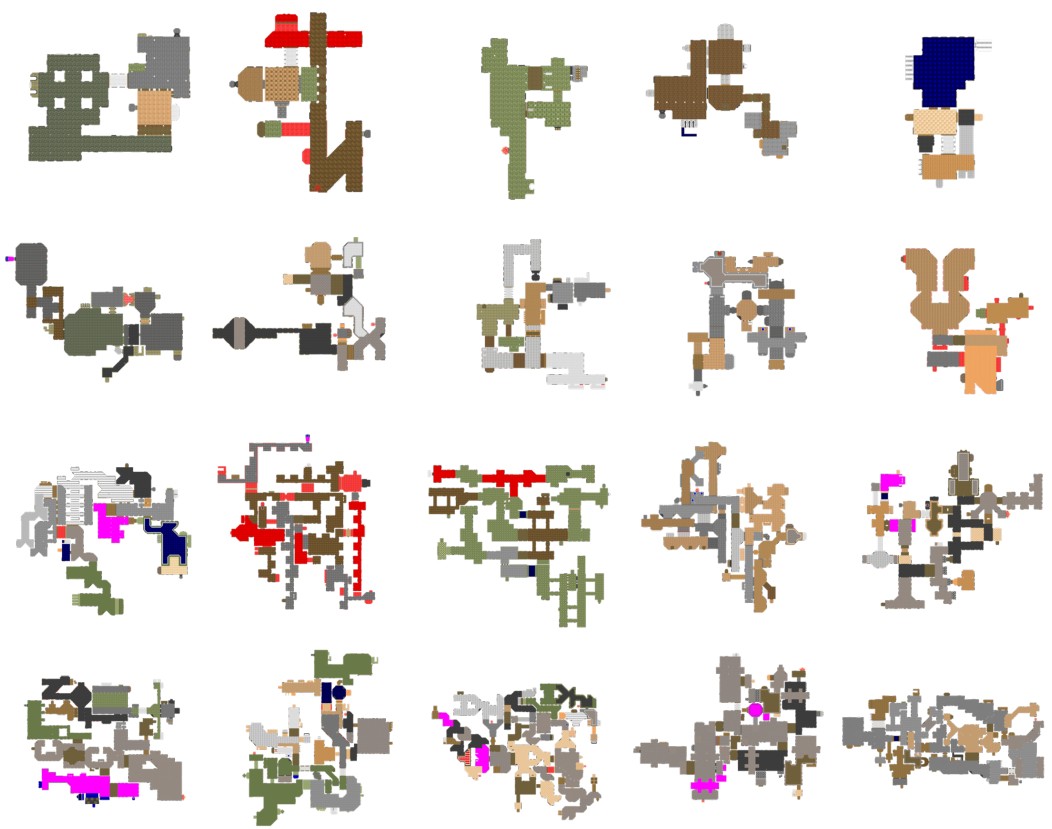

Figure 6: **More maps from our dataset.** Rows from top to bottom represent increasing scene complexity, categorized into four levels: Simple, Normal, Hard, Insane.

In addition, we apply a similar approach to project the camera's historical trajectory, resulting in a single 2D image. We filter the camera's historical positions based on their proximity to the current camera position in the XZ-plane, using the same threshold $\tau_{xz}$:

$$\mathcal{C}_t^f = \{c_k^{pos} = (x_k, y_k, z_k) \mid k < t, |x_k - x_t| \leq \tau_{xz} \text{ and } |z_k - z_t| \leq \tau_{xz}\}. \tag{10}$$

We then map these filtered positions onto a single image $H_{c_t}$ of the same size $H \times W$:

$$H_{c_t}(u, v) = \sum_{c_k^{pos} \in \mathcal{C}_t^f} \delta(\phi(c_k^{pos}) - (u, v)) \tag{11}$$

This results in a single-density image $H_{c_t}$ representing the camera's historical trajectory near its current position. To synthesize the information obtained, we define a set $\mathcal{E}_{c_t}$ encapsulating the entirety of the current exploration embedding: $\mathcal{E}_{c_t} = \{I_{c_t,1}, \ldots, I_{c_t,n}, H_{c_t}\}$.

## C   EXPERIMENTS

**Detailed quantitative results.** Table 7 and Table 8 show our superior performance on both the AiMDoom training set and the test set. Furthermore, we offer detailed results for each test scene in MP3D, as illustrated in Table 6.

**Additional ablation study.** We study the impact of different spatial range information used to predict the next best path by training four different models on the AiMDoom Normal level training split. These models processed input crop sizes ranging from $20m \times 20m$ to $50m \times 50m$, with each model tasked with predicting a value map and an obstacle map within a $40m \times 40m$ area. The Table 9 shows the results.

Table 6: Evaluation results for each test scene on MP3D dataset.

| Scene | Rooms | Comp. (%) ↑ | | | | | | Comp. (cm) ↓ | | | | | |
|---|---|---|---|---|---|---|---|---|---|---|---|---|---|
| | | Random | FBE | UPEN | OccAnt | ANM | NBP (ours) | Random | FBE | UPEN | OccAnt | ANM | NBP (ours) |
| GdvgF* | 6 | 68.45 | 81.78 | 82.39 | 80.24 | 80.99 | **87.80** | 11.67 | 5.48 | 5.14 | 5.66 | 5.69 | **4.92** |
| gZ6f7 | 1 | 29.81 | 81.01 | 82.96 | 82.02 | 80.68 | **89.91** | 46.48 | 7.06 | 6.14 | 6.19 | 7.43 | **3.31** |
| HxpKQ* | 8 | 46.93 | 58.71 | 52.70 | 60.50 | 48.34 | **66.28** | 19.10 | 11.75 | 14.11 | 11.75 | 15.96 | **8.12** |
| pLe4w | 2 | 32.92 | 66.09 | 66.76 | 67.13 | **76.41** | 71.34 | 30.79 | 12.78 | 11.82 | 11.51 | **8.03** | 9.53 |
| YmJkq | 4 | 50.26 | 68.32 | 60.47 | 68.70 | 79.35 | **81.57** | 24.61 | 11.85 | 15.77 | 11.90 | 8.46 | **8.01** |
| mean | 4 | 45.67 | 71.18 | 69.06 | 71.72 | 73.15 | **79.38** | 26.53 | 9.78 | 10.60 | 9.40 | 9.11 | **6.78** |

Table 7: Evaluation results on AiMDoom dataset (Simple and Normal).

| | AiMDoom Simple | | | | AiMDoom Normal | | | |
|---|---|---|---|---|---|---|---|---|
| | Seen | | Unseen | | Seen | | Unseen | |
| | Final Cov. | AUC | Final Cov. | AUC | Final Cov. | AUC | Final Cov. | AUC |
| Random Walk | 0.362 | 0.306 | 0.323 | 0.270 | 0.198 | 0.159 | 0.190 | 0.152 |
| | ±0.175 | ±0.156 | ±0.156 | ±0.135 | ±0.125 | ±0.104 | ±0.124 | ±0.103 |
| FBE | 0.770 | 0.628 | 0.760 | 0.605 | 0.564 | 0.423 | 0.565 | 0.415 |
| | ±0.163 | ±0.147 | ±0.174 | ±0.171 | ±0.171 | ±0.127 | ±0.139 | ±0.109 |
| SCONE | 0.597 | 0.482 | 0.577 | 0.483 | 0.421 | 0.315 | 0.412 | 0.313 |
| | ±0.177 | ±0.158 | ±0.173 | ±0.138 | ±0.138 | ±0.102 | ±0.114 | ±0.087 |
| MACARONS | 0.600 | 0.483 | 0.599 | 0.479 | 0.442 | 0.332 | 0.418 | 0.314 |
| | ±0.176 | ±0.145 | ±0.200 | ±0.172 | ±0.135 | ±0.104 | ±0.120 | ±0.088 |
| **NBP (Ours)** | **0.870** | **0.697** | **0.879** | **0.692** | **0.746** | **0.538** | **0.734** | **0.526** |
| | ±0.121 | ±0.134 | ±0.142 | ±0.156 | ±0.152 | ±0.142 | ±0.142 | ±0.112 |

Table 8: Evaluation results on AiMDoom dataset (Hard and Insane).

| | AiMDoom Hard | | | | AiMDoom Insane | | | |
|---|---|---|---|---|---|---|---|---|
| | Seen | | Unseen | | Seen | | Unseen | |
| | Final Cov. | AUC | Final Cov. | AUC | Final Cov. | AUC | Final Cov. | AUC |
| Random Walk | 0.121 | 0.086 | 0.124 | 0.088 | 0.070 | 0.048 | 0.074 | 0.050 |
| | ±0.081 | ±0.062 | ±0.082 | ±0.060 | ±0.049 | ±0.038 | ±0.048 | ±0.035 |
| FBE | 0.426 | 0.310 | 0.425 | 0.311 | 0.313 | 0.226 | 0.330 | 0.239 |
| | ±0.119 | ±0.091 | ±0.114 | ±0.080 | ±0.082 | ±0.066 | ±0.097 | ±0.079 |
| SCONE | 0.271 | 0.199 | 0.290 | 0.210 | 0.204 | 0.146 | 0.196 | 0.140 |
| | ±0.100 | ±0.172 | ±0.093 | ±0.072 | ±0.069 | ±0.052 | ±0.079 | ±0.060 |
| MACARONS | 0.316 | 0.202 | 0.302 | 0.218 | 0.201 | 0.143 | 0.192 | 0.139 |
| | ±0.106 | ±0.074 | ±0.097 | ±0.070 | ±0.068 | ±0.051 | ±0.078 | ±0.058 |
| **NBP (Ours)** | **0.627** | **0.430** | **0.618** | **0.432** | **0.486** | **0.315** | **0.472** | **0.312** |
| | ±0.144 | ±0.111 | ±0.153 | ±0.115 | ±0.106 | ±0.047 | ±0.095 | ±0.073 |

The results indicate that optimal performance is achieved when the input crop size corresponds to the crop size of the area being predicted. This is due to the fact that when the input crop size is either smaller or larger than that of the output maps, predictive errors arise. Specifically, if the input crop size is too small, it limits the model's ability to formulate effective long-term objectives. Conversely, when the input crop size is too large, the predictions for obstacles near the camera become less accurate, negatively impacting both exploration and reconstruction efficiency.

We also investigate the different strategies in inference. We conducted this experiment on the AiM-Doom Normal level, extending our previous ablation studies. Table 10 shows the results, the Original Strategy adheres to the original approach of updating long-term goals upon completing a path, while the New strategy updates goals at each step.

The results indicate that the New Strategy, which frequently updates long-term goals, performs worse than the Original Strategy. This inferior performance is mainly due to the lack of decision

Table 9: Comparison of different spatial ranges of information used to predict the next best path.

| Range | 20m × 20m | 30m × 30m | 40m × 40m | 50m × 50m |
|---|---|---|---|---|
| Final Cov. | 0.630 ±0.151 | 0.691 ±0.140 | **0.734** ±0.142 | 0.647 ±0.144 |
| AUCs | 0.469 ±0.107 | 0.501 ±0.106 | **0.526** ±0.112 | 0.457 ±0.106 |

continuity in the New Strategy, where the agent frequently changes its long-term goals. Such frequent shifts can cause the agent to oscillate between paths, wasting movement steps, particularly as our experiments were conducted with a limited number of steps. Additionally, the predictive accuracy of the value map is not perfect, and forecasting over long distances naturally entails uncertainty. New Strategy accumulates more predictive errors by recalculating predictions at every step, and frequent updates in decision-making can exacerbate these errors.

Table 10: Comparison of Different Strategies

| Strategy | Final Cov. | AUCs |
|---|---|---|
| Original strategy | **0.734** ±0.142 | **0.526** ±0.112 |
| New strategy | 0.432 ±0.168 | 0.367 ±0.135 |

Despite these challenges, our results still surpassed the performance of previous state-of-the-art next-best-view (NBV) methods, as detailed in Table. 2. This suggests that predicting coverage gains over long distances can indeed benefit efficient active mapping, even when the goal is updated at each step.

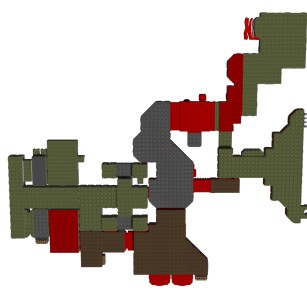
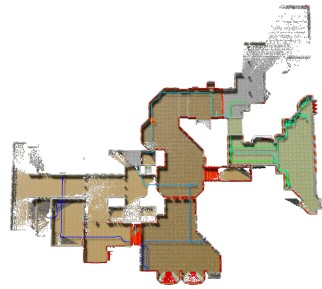

(a) Ground truth mesh           (b) NBP (Ours)

Figure 7: **Failure case 1**: Our method initially prioritizes the exploration of high-value areas, inadvertently neglecting regions of secondary importance. Thus, it results in incomplete reconstruction in the initial area of the beginning trajectory.

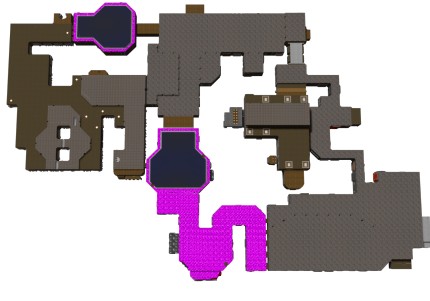
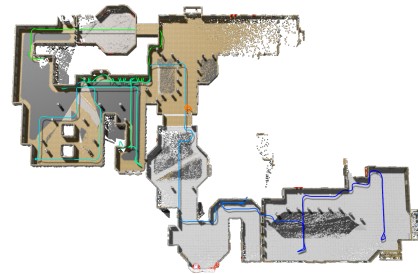

(a) Ground truth mesh           (b) NBP (Ours)

Figure 8: **Failure case 2**: This scene contains multiple narrow areas, prompting our method to depend more heavily on our precise prediction of obstacles. Under these challenging conditions, our approach may overlook exploring this area.

**Failure cases.** As Figure. 7 and Figure. 8 illustrated, we also show that in very complex environments, we could only achieve about 65% coverage. This is because, in complex environments, our method prioritizes the exploration of areas with multiple valuable goals, ignoring places of lesser

current value. After the initial exploration is complete, it is likely to explore other regions, overlooking previously encountered areas with higher value. Consequently, developing methods that aim to achieve a global optimum is a promising and valuable direction for future research.

