# OpenReview forum: "NextBestPath: Efficient 3D Mapping of Unseen Environments"
_ICLR.cc/2025/Conference — ICLR 2025 Poster_

### Official Review · Reviewer_Ayjg · 2024-10-28

**Soundness:** 2
**Presentation:** 2
**Contribution:** 2
**Rating:** 3
**Confidence:** 3

**Summary:**

This paper proposes a path-planning algorithm for efficient 3D mapping. The core components of the NBP are the coverage gain decoder and obstacle map decoder, which are leveraged for long-term goal selection. The experiments showcase significant improvements against baselines. Besides, this paper also proposes a  new dataset called AiMDoom, which includes 400 scenes with different level of diffculty.

**Strengths:**

Developing methods for highly efficient exploration is an intriguing topic with potential applications across various navigation tasks. This paper provides comprehensive details on NPV techniques, and the dataset is expected to benefit the community by supporting further investigation into exploration strategies.

**Weaknesses:**

- The scenes in AiMDoom contain minimal furniture or objects, resulting in mostly open space. This does not align with real-world environments, making these scenes suboptimal for training and evaluation purposes.

- The states described in papers L199 and L373 indicate that the proposed method operates within a 3-DoF domain. However, NBV tasks often involve planning in a 6-DoF camera pose space. Moreover, baseline methods, such as MACARONS and SCONE, support 6-DoF camera pose planning. A discussion is needed to explain why this paper considers only a 3-DoF setting.

- For 3-DoF trajectory planning, several well-known works exist, such as [1]. The authors should discuss why this paper’s approach offers advantages over previous works.

[1] TARE: A Hierarchical Framework for Efficiently ExploringComplex 3D Environments

**Questions:**

- Do the obstacle map and value map encompass the entire scene? This could result in significant computational costs in large-scale environments.

- If the long-term goal is updated at each step, does this strategy enhance performance?

- In Table 2, `comp.` represents the average minimum distance between ground truth vertices and observations. However, the HM3D depth data should be noise-free. If I haven't overlooked any details, why is the `comp.` metric considered valid?

---

> ### Author Response · Authors · 2024-11-18
> **Response to Reviewer Ayjg**
>
> Thank you for your comments. We address each point in the following.
>
> > W1: The scenes in AiMDoom contain minimal furniture or objects, resulting in mostly open space. This does not align with real-world environments, making these scenes suboptimal for training and evaluation purposes.
>
> Existing methods face challenges in actively mapping large and complex 3D scenes. The proposed AiMDoom dataset mainly aims to provide a systematic benchmark for evaluating models’ capabilities across various difficulty levels of scenes for active mapping. As shown in Table 1, AiMDoom has greater navigation complexity than existing real-world datasets. It is also easier to scale up in dataset size with the automatic generation code.
>
> > W2: A discussion is needed to explain why this paper considers only a 3-DoF setting, while baseline methods, such as MACARONS and SCONE, support 6-DoF camera pose planning.
>
> > W3: For 3-DoF trajectory planning, several well-known works exist. The authors should discuss why this paper’s approach offers advantages over previous works.
>
> Though baseline NBV methods for single objects or outdoor scenes use a 6-DoF setting, existing methods for indoor 3D mapping such as those evaluated in the MP3D dataset [1,2,3], commonly utilize a 3-DoF setting with actions limited to turning left, turning right, and moving forward. In alignment with these prior works and the focus on indoor 3D scenes, we keep the 3-DoF setting as in previous work.
>
> In Table 3 we already do compare against several recent works for 3-DoF trajectory planning. Thank you for mentioning the TARE paper on 3-DoF trajectory planning. TARE employs a hierarchical strategy for exploration based on non-learning control and planning optimization; its viewpoint sampling process in subspace is confined to the sensor's range, where they use additional lidar sensors. In contrast, our proposed learning-based method can predict optimal poses and potential obstacles over a broader range even using only a single depth sensor. Due to differences in focus areas of the tasks and settings in simulators, we did not compare against TARE methods in our experiments. We will include discussions of these studies in the related work section.
>
> > Q1: Do the obstacle map and value map encompass the entire scene? This could result in significant computational costs in large-scale environments.
>
> We crop the scene centered around the current camera position for obstacle map and value map prediction. We convert the 3D point cloud into a stack of 2D images as inputs, which makes it more scalable to large environments. You could refer to Section 2 of the supplementary material for the details.
>
> > Q2: If the long-term goal is updated at each step, does this strategy enhance performance?
>
> Thank you for the suggestion. It is interesting to explore. We will conduct further experiments on this issue in the coming days.
>
> > Q3: In Table 2, why is the comp. metric considered valid?
>
> This metric evaluates the completeness of the reconstruction. Since all the methods use the same maximum budget for exploration, the comp(%) metric will be low if a model fails to explore some areas within the budget. Therefore, existing methods on the MP3D dataset all use this metric.
>
> [1] Active neural mapping, ICCV 2023
>
> [2] Occupancy anticipation for efficient exploration and navigation, ECCV 2022
>
> [3] Uncertainty-driven planner for exploration and navigation, ICRA 2022

---

> ### Author Response · Authors · 2024-11-21
> **Update on the experiment results in Q2**
>
> We conducted this experiment on the AiMDoom Normal level, extending our previous ablation studies. The table below shows the results, the Original Strategy adheres to the original approach of updating long-term goals upon completing a path, while the New strategy updates goals at each step.
>
> | Strategy | Final Cov. | AUCs |
> |:--|:--:|:--:|
> | Original strategy | **0.734**(±0.142) | **0.526**(±0.112) |
> | New strategy | 0.432(±0.168) | 0.367(±0.135) |
>
> The results indicate that the New Strategy, which frequently updates long-term goals, performs worse than the Original Strategy. This inferior performance is mainly due to the lack of decision continuity in the New Strategy, where the agent frequently changes its long-term goals. Such frequent shifts can cause the agent to oscillate between paths, wasting movement steps, particularly as our experiments were conducted with a limited number of steps. Additionally, the predictive accuracy of the value map is not perfect, and forecasting over long distances naturally entails uncertainty. New Strategy accumulates more predictive errors by recalculating predictions at every step, and frequent updates in decision-making can exacerbate these errors.
>
> Despite these challenges, our results still surpassed the performance of previous state-of-the-art next-best-view (NBV) methods, as detailed in Table 2 of the main paper. This suggests that predicting coverage gains over long distances can indeed benefit efficient active mapping, even when the goal is updated at each step.

---

> > ### Comment · Reviewer_Ayjg · 2024-12-02
> >
> > Thank you for the comprehensive response, which addresses most of my concerns.
> >
> > However, I still find that the proposed environment deviates significantly from real-world settings. It is largely empty, which contrasts with the crowded nature of typical indoor scenes.
> >
> > This leads to another issue: the planning trajectories tend to be overly simplistic (e.g., a straight line across a single room). I had anticipated more challenging scenarios, such as planning a trajectory that efficiently explores all objects within the room.

---

> > > ### Author Response · Authors · 2024-12-02
> > >
> > > Thank you for your feedback. We are glad to have addressed most of your concerns. We address your remaining concerns in the following.
> > >
> > > > The proposed environment is largely empty.
> > >
> > > Our AiMDoom dataset is motivated by a project on digital twins of construction sites of buildings, an application with a huge market. In such conditions, the environments are large, with many rooms and small openings, but not many objects. Our AiMDoom dataset is representative of the challenges raised by such environments.
> > >
> > > Also, while there are not many objects in our AiMDoom environments, these environments are objectively more complex than earlier datasets according to the navigation complexity metric in Table 1.
> > >
> > > > The planning trajectories tend to be overly simplistic.
> > >
> > > Due to the characteristic of our AiMDoom environments, the learned trajectories e.g. straight lines are efficient when one wants to minimize the time required for mapping.
> > >
> > > Please note that we do evaluate our approach on environments from MP3D that have many objects. Please see Table 3 in the main paper and Table 1 in the supplementary material for further details. We cannot submit a revision at this time, but we will add figures of our trajectories on MP3D.
> > >
> > > Our experimental results demonstrate that the proposed approach can generalize to very different environments with few or many objects.

---

### Official Review · Reviewer_aPVh · 2024-10-29

**Soundness:** 3
**Presentation:** 2
**Contribution:** 2
**Rating:** 6
**Confidence:** 5

**Summary:**

This paper proposes a learning-based method to the problem of active 3D mapping of unknown environments. The method is hinged on next-best-path (NBP). It integrates a mapping progress encoder, a coverage gain decoder and an obstacle map decoder. The coverage gain and the obstacle map are used to compute the NBP. The NBP can direct the robot to reconstruct unseen environments with predicted long-term goals, achieving state-of-the-art performance on both the MP3D and AiMDoom datasets. The paper also contributes a dataset, AiMDoom, designed to benchmark active mapping in indoor scenes.

**Strengths:**

The idea of estimating NBP has good merit and shows promising results. The method is overall reasonably designed and the results are good. The evaluation demonstrates the strength of the proposed method.

**Weaknesses:**

The paper claims that the main novelty is the idea of next best path planning. It shows that NBP performs better than NBV which is reasonable and convincing. However, the method how NBP is computed is rather simplistic and the major technical components are actually the reconstructed map encoder and the two map decoders. With the estimated value map and obstacle map, NBP is computed in a straightforward way. On the other hand, training a network to predict value maps for scene coverage has good merit.

Some of the paths computed by the proposed method are too close to the walls, see Fig 4(b) left, making them collision-prone. The visualization of paths does not have to show shadows; showing the path on the ground could be clearer.

The experiment of spatial range of long-term goal is interesting. However, it would be more useful if more test can be conducted to find out how to choose the size of range for a given scene.

The proposed method is trained on the train split of AiMDoom. It is unclear whether the other alternative methods being compared were trained on AiMDoom or pretrained on other datasets. If so, the comparison would be unfair. More explanation is needed.

**Questions:**

No.

**Details Of Ethics Concerns:**

No.

---

> ### Author Response · Authors · 2024-11-18
> **Response to Reviewer aPVh**
>
> Thank you for your comments and for recognizing that our model is well-designed and the results are good.
>
> > W1: The networks for obstacle and value maps prediction are simple, and using networks to predict value maps for guided exploration is not new. The technical contribution of this work is weak.
>
> Our main technical contributions lie beyond the network design.
>
> First, we propose a new paradigm by shifting from state-of-the-art next-best-view prediction to next-best-path prediction for active mapping.
>
> Second, compared to prior works that also predict long-term goals, we propose a new criterion for selecting long-term goals based on coverage gains, which is more closely aligned with the final objective of active mapping. We also propose an efficient data collection method and training strategy for training the coverage gain decoder.
>
> Finally, we unify the models for obstacle prediction and value map prediction, while prior works typically use separate models for navigation and exploration goal prediction. Our unified model is more efficient and the multi-task learning further enhances the performance.
>
> We will make these contributions more clear in the revised version.
>
> > W2: Some of the paths computed by the proposed method are too close to the walls, see Fig 4(b) left, making them collision-prone. The visualization of paths does not have to show shadows.
>
> The visualization may make the trajectory look like it is close to the wall, but this is only due to the scaling and perspective of the cameras, not actual proximity.
>
> After testing, the minimum distance to the closest obstacle along our predicted trajectories is 0.6 meters for the trajectory in Fig 4(b), which ensures no risk of collision. Additionally, collision checking is implemented in our simulator to verify the validity of the path.
>
> We include the shadowing in the images to highlight that this is a 3D reconstruction task, not a 2D task. For better visualization, we included a video in the supplementary material, which more clearly demonstrates the active mapping process of our method.
>
> > W3: It would be more useful if more test can be conducted to find out how to choose the size of range for a given scene.
>
> Thank you for your suggestion. This is an interesting future direction to explore. Our experiments in Figure 5 show that the range of 40m achieves the best performance on average for all the scenes. To further investigate, we analyzed whether this range performs best for each individual scene, and found that 76.67% of the evaluated scenes achieve their best performance with this range. This result suggests that this hyper-parameter can generalize effectively across different scenes.
>
> > W4: It is unclear whether the other alternative methods being compared were trained on AiMDoom or pretrained on other datasets. If so, the comparison would be unfair.
>
> We did retrain all the methods we compared in Table 2 on the AiMDoom dataset for a fair comparison. We will make this clear in the paper. We will also release these model weights upon acceptance.

---

> > ### Comment · Reviewer_aPVh · 2024-11-27
> >
> > Thank you for the response. I recognize that the idea NBP is a contribution on the conceptual side. My main concern is the proposed method for computing NBP is quite simplistic on the technical side as learning to predict value maps is not new in the field.

---

> > > ### Author Response · Authors · 2024-11-27
> > >
> > > Thank you for acknowledging our main contribution.
> > >
> > > We do not understand the concern raised by the reviewer. It is true that earlier methods predict a map of values, but what really matters is what the values represent. All previous methods predict a value per location corresponding to the location itself; we propose to predict a value corresponding to the path from the current position of the agent to the location. This is a fundamental difference, which significantly improves the performances as our experiments show. Moreover, we provide an algorithm to learn to predict these values.
> > >
> > > If the reviewer references the architecture we use to predict these values, we believe it is important to keep it as simple as possible. Isn’t it better if a simple architecture can be used?

---

> > > > ### Comment · Reviewer_aPVh · 2024-11-28
> > > >
> > > > I meant the computation of NBP from the value maps contains no technical contribution. The value map is related to scene coverage which is indeed different from those for goal-directed navigation, which does have merit.

---

> > > > > ### Author Response · Authors · 2024-11-29
> > > > >
> > > > > Thank you very much for your further feedback and for increasing your score.
> > > > >
> > > > > Currently, we use Dijkstra’s algorithm to identify the “next best path”, and we did show this already provides excellent results. It is possible that there are better options to define the path, and this opens doors for future research.

---

### Official Review · Reviewer_E6rF · 2024-11-03

**Soundness:** 3
**Presentation:** 3
**Contribution:** 3
**Rating:** 6
**Confidence:** 4

**Summary:**

In this paper, the authors focus on improving the reconstruction efficiency of active mapping in a new environment. Previous methods mainly predict the next best viewpoint near the current location and are prone to getting stuck in local areas.

Instead, the authors propose the leverage accumulated history information to find a long-term goal which could bring the largest gain. At the same time, an obstacle map is predicted to find the long-term goal efficiently.

In addition to the proposed method, the authors also introduce a new synthetic dataset which has more complicated structures and larger diversities of map than previous datasets.

Experiments on one public dataset and the new dataset demonstrate that the proposed method outperforms prior methods significantly.

**Strengths:**

The strengths of this paper are as follows:

1.	A more complicated dataset (AiMDoom) for active mapping. Compared with other either synthetic or real datasets, such as Replica, RoboTHOR, MP3D, and ScanNet, HM3D, the new dataset AiMDoom has more scenes, larger area size and different levels of difficulty. Intricate geometries and layouts, small doors and narrow corridors, the high diversity of scenes bring new challenges to the active mapping task. This benefits the whole community.

2.	A novel approach for active mapping. The authors propose to predict the next best path (NBP) to find the next optimal location instead of directly predicting the one close to current position of the agent. The value map in NBP provides the best location and the obstacle map allows to use Dijkstra algorithm to find the shortest path from current location to the goal. This is a useful combination and brings inspiration to following works.

3.	Impressive performance. The proposed approach obtains state-of-the-art performance on MP3D dataset and gives significantly better results than previous methods like MACARONS on the new dataset. Ablation studies show the effectiveness of the obstacle map.

4.	The paper is well-organized and easy to read.

**Weaknesses:**

I don’t see obvious weaknesses of this paper. Some of my concerns about the method and the dataset are as follows.

1. In L243, the point clouds are cropped at the current location of the agent. I am curious how does it work and what kind of parameters are used? My understanding is that the crop size may influence how much history information is used for the next path prediction.

2. In L245, the 3D point clouds are projected onto 2D image to simplify the processing. This strategy works for scenes with a single layer but may lose generalization ability in scenes with multiple layers as part of the depth information is discarded.

3. In the ablation study, the efficacy to the final reconstruction results of both the obstacle map and multi-task training are tested, however, it would be good to see the accuracy of the obstacle map itself and the value map itself instead of final reconstruction accuracy.

4. The new dataset is a synthetic dataset, so the domain gap may exist. As far as I know, a new version of Scannet dataset has been released. It has more scenes with more complicated geometry and structures. In the future, it might be possible to reorganize this new Scannet dataset and make a real complicated dataset.

**Questions:**

See the weaknesses section.

---

> ### Author Response · Authors · 2024-11-18
> **Response to Reviewer E6rF**
>
> We thank the reviewer for providing constructive comments and the recognition of our work.
>
> > Q1: In L243, the point clouds are cropped at the current location of the agent. How does it work and what kind of parameters are used? My understanding is that the crop size may influence how much history information is used for the next path prediction.
>
> The point cloud is sliced vertically based on the camera's position and transformed into density images through a projection function that normalizes point coordinates into image space. Key parameters in this transformation include the camera's current position, which centers the projection, and the radius (40m$\times$40m), which defines the observational area around the camera. The image dimensions, set at 256$\times$256, determine the resolution of the output images. This projection function adjusts the 3D coordinates to align within these dimensions, ensuring that the value of each pixel in the resulting images accurately represents the density of the point cloud data.
>
> You could refer to Section 2 of the supplementary material for a clearer understanding of these details and the formulas.
>
> The larger the crop size, the more history information is used for the next path prediction.  We will provide an ablation on the crop size for your interest in the next few days.
>
> > Q2: In L245, the 3D point clouds are projected onto 2D image to simplify the processing. This strategy works for scenes with a single layer but may lose generalization ability in scenes with multiple layers as part of the depth information is discarded.
>
> We use a stack of 2D images rather than a single image to represent the scene, which has the potential to capture essential depth information even for scenes with multiple layers. Since prior works focus on single-layer scenes and it is already challenging to reconstruct large single-layer scenes, we do not prioritize results for multi-layer scenes in this work.
>
> > Q3: In the ablation study, the efficacy to the final reconstruction results of both the obstacle map and multi-task training are tested, however, it would be good to see the accuracy of the obstacle map itself and the value map itself instead of final reconstruction accuracy.
>
> Thank you for the suggestion. For the prediction of the value map, it is prohibitively expensive to obtain ground-truth coverage gains for all pixels in the map. This is why we use per-pixel data for training value maps, and thus we cannot report the performance of value map prediction in inference.
>
> For the obstacle map, we can evaluate its performance given the ground-truth obstacle map. The table below presents our experimental results. The results from the ablation study continue to demonstrate that multi-task learning outperforms training tasks independently, indicating that the tasks are mutually beneficial and collectively enhance learning.
> | Strategy | Accuracy | Precision|
> |:---------|:--------|:-----------|
> | Single-task | 0.968| 0.754|
> | Multi-task | **0.970**| **0.805**|
>
> > Q4: The new dataset is synthetic, which may create a domain gap. A more recent version of the Scannet dataset could be used in the future.
>
> Thank you for your feedback. Considering ScanNet++ is certainly interesting, however our dataset is designed to address the challenge of actively reconstructing large-scale complex 3D environments, which is one of the key difficulties for existing methods. As indicated in Table 1, real-world datasets including the Scannet are limited by smaller navigation spaces and lower complexity. The ScanNet++ you mentioned does not improve upon these aspects. We also plan to release our codes and models, enabling future research to evaluate our methods on other datasets like Scannet++.

---

> ### Author Response · Authors · 2024-11-21
> **Update on the experiment results in Q1**
>
> We conducted this ablation study on crop size, training four different models on the AiMDoom Normal level training split. These models processed input crop sizes ranging from 20m × 20m to 50m × 50m, with each model tasked with predicting a value map and an obstacle map within a 40m × 40m area. The table below shows the results.
>
> | Range | 20m × 20m | 30m × 30m | 40m × 40m | 50m × 50m |
> |:--|:--:|:--:|:--:|:--:|
> | Final Cov. | 0.630(±0.151) | 0.691(±0.140) | **0.734**(±0.142) | 0.647(±0.144) |
> | AUCs | 0.469(±0.107) | 0.501(±0.106) | **0.526**(±0.112) | 0.457(±0.106) |
>
> The results indicate that the best results are achieved when the input crop size matches the crop size of the area being predicted. This is because when the input crop size is either smaller or larger than the crop size of the output maps, it leads to predictive errors. If the input crop size is too small, it restricts the model’s ability to formulate effective long-term goals. Conversely, if the input crop size is too large, the predictions for obstacles near the camera become inaccurate, adversely affecting both exploration and reconstruction efficiency.

---

### Official Review · Reviewer_1qFw · 2024-11-04

**Soundness:** 3
**Presentation:** 3
**Contribution:** 2
**Rating:** 5
**Confidence:** 5

**Summary:**

This paper proposes a method to predict paths during mapping that optimize accumulated coverage. The goal is to cover the environment in a minimal time using only a depth sensor.

Point clouds and the robot trajectory are input to an encoder which yields a latent that is decoded into a value map representing the coverage gain and an occupancy map. The cell in the coverage gain map with the highest value is set as the long term goal.

The authors proposed a new dataset AiMDoom played out on a game (simulation) environment, since active methods unlike passive SLAM have to be evaluated on the same environment with different actions/trajectories.

**Strengths:**

S1: the new dataset has complex challenging lay-outs.

S2:  the new dataset has diversity with a lot of opportunities to evaluate generalization

S3: the decoded occupancy map includes unseen places, behind walls, for example.

**Weaknesses:**

W1: The main weakness of the paper is the scope of active mapping addressed is only coverage rather than the map itself. While coverage is indeed significant it assumes that  3D reconstructions are error-free (the M in SLAM). Moreover, poses are assumed accurate, an assumption far from reality.

W2: The paper is set in a very narrow context by ignoring the literature on Active SLAM. In particular, active mapping has been based on first principles of information theory. See the excellent exposition here Julio A Placed, Jared Strader, Henry Carrillo, Nikolay Atanasov, Vadim Indelman, Luca Carlone,
and José A Castellanos. A survey on active simultaneous localization and mapping: State of the art and new frontiers. IEEE Transactions on Robotics, 2023

I think the authors would benefit a lot in rethinking their approach and rewriting their paper by reading this article.

W3: The approach is very similar to (Georgakis, 2022). While Georgakis et al. predict occupancy probability and model uncertainty, here the authors predict occupancy and a value map that should have the interpretation of information gain/uncertainty. While Georgakis' objective is point-goal navigation one can use its exploration policy as a pure mapper. Georgakis' value map is based on explicit computation of covariance from ensembles without the use of any ground-truth.
Finally, Georgakis choose long-term goal and then estimate paths based on occupancy maps, similar to the approach here.

W4: The main idea of exploration is trying to choose paths where the measurements are not predictable by the occupancy paths. The expression in (2), however, defines the gain as minimal error to the ground-truth. This will not encourage the agent to go to new unvisited directions but rather to directions where the prediction error will be very small.

W5: There is considerable literature that has been ignored in related work and experimental comparisons. In particular, we would like to see comparisons with

a.  D. S. Chaplot, D. Gandhi, S. Gupta, A. Gupta, and R. Salakhutdinov, “Learning to explore using active neural SLAM,” in Proc. Int. Conf. Learn. Representations, 2020.

b. A. Bircher, M. Kamel, K. Alexis, H. Oleynikova, and R. Siegwart, “Recedinghorizon“next-best-view”plannerfor3Dexploration,”inProc. IEEE Int. Conf. Robot. Autom., 2016, pp. 1462–1468.

**Questions:**

Q1. L068: You write "scene uncertainty does not directly align with the ultimate objective of 3D mapping". Is the uncertainty of predicted occupancy not the uncertainty of the 3D map? What do you mean here?

Q2: The computed information gain during training is using the ground-truth ( eq 2) while inference uses the ground-truth instead. It is not clear whether the use of the ground-truth

Q3: The authors should clarify the first term in eq. 3. What does it mean ground-truth coverage gain?

---

> ### Author Response · Authors · 2024-11-18
> **Response to Reviewer 1qFw (Part 1)**
>
> We thank the reviewer for the detailed comments. We address the raised points in the following.
>
> >W1: The active mapping task addressed in the paper focuses solely on maximizing coverage and relies on the assumption of accurate poses.
>
> >W2: The paper is set in a very narrow context and lacks discussion with active SLAM.
>
> >W5: We would like to see comparisons with active SLAM methods (Chaplot et al., ICLR 2020) and (Bircher et al., ICRA 2016).
>
> It is true that our work focuses on maximizing coverage and assumes accurate poses, however, this is also true for numerous earlier work which we reference in [1-8]. This focus allows the development of algorithms capable of reconstructing detailed 3D models of complex environments while minimizing exploration time, which is still a very challenging task.
>
> Please also note that we outperform UPEN [9] by a large margin, while UPEN already achieved higher map coverage than (Chaplot et al., ICLR 2020) on the MP3D dataset (67.9 % vs. 52.1%). In the case of Bircher et al. (ICRA 2016), they consider sensors and a task setup that differ significantly from ours, making a direct comparison infeasible.
>
> This being said, we agree we should have emphasized the difference between active mapping as in [1-8] and active SLAM, and we will clarify the active mapping problem we addressed in the paper.
>
> > Q1. L068: You write "scene uncertainty does not directly align with the ultimate objective of 3D mapping". Is the uncertainty of predicted occupancy not the uncertainty of the 3D map? What do you mean here?
>
> We aim to maximize the coverage of the scene by the camera, because it was shown to be a good criterion for active mapping in [1-8]. Maximizing coverage is related to minimizing uncertainty, but maximizing coverage formalizes better the ultimate goal of active mapping (Please also see our answer above about the emphasis on active mapping).
>
> > Q2: The computed information gain during training is using the ground-truth (eq 2) while inference uses the ground-truth instead. It is not clear whether the use of the ground-truth.
>
> During training, we use the ground-truth point cloud to calculate the coverage gain when moving the camera from pose A to pose B along a trajectory. This coverage gain is used to train the coverage gain decoder at camera pose A.
>
> In the inference phase, our model automatically predicts the coverage gain in the value map, and uses it to select the next best path. The ground-truth is only used in evaluation to calculate the AUCs metrics.
>
> > Q3: The authors should clarify the first term in eq. 3. What does it mean ground-truth coverage gain?
>
> As replied in Q2, we use the ground-truth point cloud to calculate the coverage gain between camera poses. This is treated as ground-truth coverage gain in Eq (3).
>
> > W3: The approach is very similar to (Georgakis, 2022).
>
> > 1) Georgakis et al. model uncertainty, here the authors predict occupancy and a value map that should have the interpretation of information gain/uncertainty;
>
> > 2) While Georgakis' objective is point-goal navigation, one can use its exploration policy as a pure mapper.
>
> We respectfully disagree with the statement that our approach is very similar to (Georgakis, 2022):
>
> * (Georgakis, 2022) considers paths as well, but relies on the average of the uncertainties at each point on paths sampled with Rapidly exploring Random Trees (RRTs). This uncertainty average is not really representative of the value of the path, as it is possible that seeing the scene from one point on the path will remove the uncertainties for the other points on the path evaluated on the path. By contrast, we learn to predict the coverage gain obtained by summing all the coverage gains obtained by moving along the path.
>
> In fact, we compared our approach to (Georgakis, 2022) on the MP3D dataset. The results, presented in Table 3 of the main paper and Table 1 of the supplementary material show that our approach outperforms (Georgakis, 2022) by a large margin, with over 10% absolute improvement in the Comp(%) metric. Note that we could not evaluate (Georgakis, 2022) on our AiMDoom dataset due to the substantial resources required to train their entire system (64 GPUs over three days of training for the pretrained navigation model they used). To ensure a fair comparison, all methods we compared in Table 2 were trained on our AiMDoom dataset.
>
> > 3) Georgakis' value map is based on explicit computation of covariance from ensembles without the use of any ground-truth.
>
> Note that we use the ground truth coverage gain only during training. This ground truth is computed automatically and does not require any human intervention. We thus do not see this as a drawback.

---

> ### Author Response · Authors · 2024-11-18
> **Response to Reviewer 1qFw (Part 2)**
>
> > 4) Georgakis chose a long-term goal and then estimated paths based on occupancy maps, similar to the approach here.
>
> Please see our answer for Point 1) for the fundamental difference between how we and (Georgakis, 2022) measure the value of a path.
>
> Moreover, their approach incorporates a pretrained navigation model which is limited to only two datasets and requires large GPU resources to train. In contrast, we directly predict an occupancy map and use Dijkstra's algorithm to generate a trajectory for navigation.
>
> > W4: The expression in (2) defines the gain as minimal error to the ground-truth. This will not encourage the agent to go to new unvisited directions but rather to directions where the prediction error will be very small.
>
> It seems there is a misunderstanding here: Equation 2 measures the difference in coverage between a new viewpoint and the current viewpoint: This encourages the agent to select viewpoints from which new parts of the scene can be seen.
>
> Additionally, to avoid falling into local optima during training, we use Equation 1 to effectively balance exploration and exploitation.
>
> [1] A reinforcement learning approach to the view planning problem, CVPR 2017
>
> [2] Next-best view policy for 3d reconstruction, ECCV-W 2020
>
> [3] GenNBV: Generalizable Next-Best-View Policy for Active 3D Reconstruction, CVPR 2024
>
> [4] Macarons: Mapping and coverage anticipation with rgb online self-supervision, CVPR 2023
>
> [5] Scone: Surface coverage optimization in unknown environments by volumetric integration, NIPS 2022
>
> [6] Active neural mapping, ICCV 2023
>
> [7] NARUTO: Neural Active Reconstruction from Uncertain Target Observations, CVPR 2024
>
> [8] Active Neural Mapping at Scale, IROS 2024
>
> [9] Uncertainty-driven planner for exploration and navigation, ICRA 2022

---

### Author Response · Authors · 2024-11-24

Dear All Reviewers,

Thank you for taking the time and effort to review our paper and for providing insightful feedback.

We have carefully addressed your comments through additional experiments, clarifications, and a revised submission. We hope these efforts have resolved your concerns and convinced you to adjust the scores. We are looking forward to your feedback and are happy to engage in further discussions if you have any remaining concerns.

Best regards,

The Authors

---

### Meta-Review · Area_Chair_8Aih · 2024-12-18

**Metareview:**

The paper investigates an interesting topic: active 3D mapping, the task of finding the shortest possible trajectory of an agent such that it reconstructs an entire scene. The agent is assumed to be equipped with a depth sensor.

The paper presents the point of view that prior art, learning-based methods that learn to estimate the next viewpoint, do not perform well in complex and cluttered scenes and that existing benchmarks are too simplistic to reveal this. To this end, the paper presents a new, more challenging benchmark, AiMDoom (rendered based on the classic Doom video game). Rendered scenarios capture complex environments.

The paper also proposes a new method, next-best-path (NBP). It takes as input point clouds and the agent trajectory and estimates (i) coverage gain and (ii) occupancy maps. The highest value in the coverage gain map is set as the long-term goal -- a location to which the next path can be estimated using classic methods, such as Dijkstra's shortest path algorithm.

Overall, the paper received mixed ratings of 6,6,5,3. Two reviewers endorse this paper, and one argues against acceptance. The reviewer that provided the final rating of 5 stated they are upgrading their rating after the discussion; however, it looks like they forgot to update the review.

Reviewers appreciate the new dataset that presents new challenges in the active mapping field and are intrigued that the proposed approach can estimate occupancy in areas not directly observed (e.g., behind walls). Reviewers also appreciate the proposed goal-oriented method, appreciate that it works well in challenging scenes, and find the paper overall well written.

However, reviewers also raise several concerns.
In particular, 1qFw initiated a discussion on the paper's scope in relation to SLAM and pointed out similarities to the prior art (Georgakis, 2022). In particular, the reviewer discusses similarities to the work by Georgakis et al. that also predict occupancy and model uncertainty and points out that the value map can also be interpreted via information gain/uncertainty.

The authors provided a detailed response, and 1qFw acknowledged that the author's feedback addressed the reviewer's concerns (the reviewer commented they would increase their rating; it appears they forgot to update their review).

Reviewer E6rF states that they do not find major weaknesses but ask for clarification on a few aspects of the proposed model, such as point cloud preprocessing, encoding, and comment on the domain gap (between Doom-style rendered and real data). The reviewer was happy with the author's response and retained their rating (6).

Reviewer's aPVh upgraded their rating to 6 after the discussion. Reviewer acknowledges that the core novelty is the high-level idea behind the next-best-path planning and comments that the approach is "reasonable and convincing"; however, they comment that the execution of this idea is "computed in rather simplistic and the major technical components are the reconstructed map encoder and the two map decoders. With the estimated value map and obstacle map, NBP is computed in a straightforward way" and concludes their justification for the rating with "on the technical side as learning to predict value maps is not new in the field".

I agree with the author's response that a simple implementation should be preferable, and the reviewer acknowledged (and others as well) that the proposed method is novel.

Finally, reviewer Ayjg argues for rejection (final rating 3), mainly because the proposed dataset (AiMDoom) is "too simplistic" and differs from real-world cluttered environments.

I do not find this argument to be sufficient to reject this paper. While Ayjg i quite right that there is a gap between utilized snyhetic datasets, and real-world data, the proposed dataset does make a step forward by providing a new dataset that covers large environments, that consists of multiple rooms, narrow passages and small openings. Paper convincingly demonstrated that prior art could not handle such scenes and proposes a method that can.

After thoroughly reading the author's feedback and discussion, I decided to side with three reviewers who (based on their comments) decided to endorse this paper. I read the paper, and agree that it challenges the status quo in active mapping, points out failure cases of prior art, and presents an intriguing new approach that addresses these shortcomings. It is, overall, a well-rounded paper.

**Additional Comments On Reviewer Discussion:**

I included comments on reviewer's discussion in my justification for the rating above.

---

### Decision · Program_Chairs · 2025-01-22

Accept (Poster)